# Learning Causal Structures Using Regression Invariance

**AmirEmad Ghassami**[*†], **Saber Salehkaleybar**[†], **Negar Kiyavash**[*†], **Kun Zhang**[‡]

[*]Department of ECE, University of Illinois at Urbana-Champaign, Urbana, USA.
[†]Coordinated Science Laboratory, University of Illinois at Urbana-Champaign, Urbana, USA.
[‡]Department of Philosophy, Carnegie Mellon University, Pittsburgh, USA.
[†]{ghassam2,sabersk,kiyavash}@illinois.edu, [‡]kunz1@cmu.edu

## Abstract

We study causal discovery in a multi-environment setting, in which the functional relations for producing the variables from their direct causes remain the same across environments, while the distribution of exogenous noises may vary. We introduce the idea of using the invariance of the functional relations of the variables to their causes across a set of environments for structure learning. We define a notion of completeness for a causal inference algorithm in this setting and prove the existence of such algorithm by proposing the baseline algorithm. Additionally, we present an alternate algorithm that has significantly improved computational and sample complexity compared to the baseline algorithm. Experiment results show that the proposed algorithm outperforms the other existing algorithms.

## 1 Introduction

Causal structure learning is a fundamental problem in machine learning with applications in multiple fields such as biology, economics, epidemiology, and computer science. When performing interventions in the system is not possible or too expensive (observation-only setting), the main approach to identifying direction of influences and learning the causal structure is to run a constraint-based or a score-based causal discovery algorithm over the data. In this case, a "complete" observational algorithm allows learning the causal structure to the extent possible, which is the Markov equivalence of the ground truth structure. When the experimenter is capable of intervening in the system to see the effect of varying one variable on the other variables (interventional setting), the causal structure could be exactly learned. In this setting, the most common identification procedure considers that the variables whose distributions have varied are the descendants of the intervened variable and hence the causal structure is reconstructed by performing interventions on different variables in the system [4, 11]. However, due to issues such as cost constraints and infeasibility of performing certain interventions, the experimenter is usually not capable of performing arbitrary interventions.

In many real-life systems, due to changes in the variables of the environment, the data generating distribution will vary over time. Considering the setup after each change as a new environment, our goal is to exploit the differences across environments to learn the underlying causal structure. In this setting, we do not intervene in the system and only use the observational data taken from environments. We consider a multi-environment setting, in which the functional relations for producing the variables from their parents remain the same across environments, while the distribution of exogenous noises may vary. Note that the standard interventional setting could be viewed as a special case of multi-environment setting in which the location and distribution of the changes across environments are designed by the experimenter. Furthermore, as will be seen in Figure 1(a), there are cases where the ordinary interventional approaches cannot take advantages of changes across environments while these changes could be utilized to learn the causal structure uniquely. The multi-environment setting was also studied in [35, 23, 37]; we will put our work into perspective in relationship to these in the Related Work.

We focus on the linear structural equation models (SEMs) with additive noise [1] as the underlying data generating model (see Section 2 for details). Note that this model is one of the most problematic models in the literature of causal inference, and if the noises follow a Gaussian distribution, for many structures, none of the existing observational approaches can identify the underlying causal structure uniquely[1]. The main idea in our proposed approach is to utilize the change of the regression coefficients, resulting from the changes across the environments, to distinguish causes from the effects.

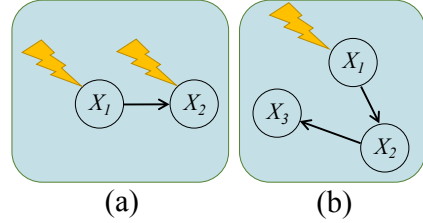

Figure 1: Simple examples of identifiable structures using the proposed approach.

Our approach is able to identify causal structures that were not identifiable using observational approaches, from information that was not usable in existing interventional approaches. Figure 1 shows two simple examples to illustrate this point. In this figure, a directed edge form variable $X_i$ to $X_j$ implies that $X_i$ is a direct cause of $X_j$, and change of an exogenous noise across environments is denoted by the flash sign. Consider the structure in Figure 1(a), with equations $X_1 = N_1$, and $X_2 = aX_1 + N_2$, where $N_1 \sim \mathcal{N}(0, \sigma_1^2)$ and $N_2 \sim \mathcal{N}(0, \sigma_2^2)$ are independent mean-zero Gaussian exogenous noises. Suppose we are interested in finding out which variable is the cause and which is the effect. We are given two environments across which the exogenous noise of both $X_1$ and $X_2$ are varied. Denoting the regression coefficient resulting from regressing $X_i$ on $X_j$ by $\beta_{X_j}(X_i)$, in this case, we have $\beta_{X_2}(X_1) = \frac{Cov(X_1, X_2)}{Cov(X_2)} = \frac{a\sigma_1^2}{a^2\sigma_1^2 + \sigma_2^2}$, and $\beta_{X_1}(X_2) = \frac{Cov(X_1, X_2)}{Cov(X_1)} = a$. Therefore, except for pathological cases for values for the variance of the exogenous noises in two environments, the regression coefficient resulting from regressing the cause variable on the effect variable varies between the two environments, while the regression coefficient from regressing the effect variable on the cause variable remains the same. Hence, the cause is distinguishable from the effect. Note that structures $X_1 \rightarrow X_2$ and $X_2 \rightarrow X_1$ are in the same Markov equivalence class and hence, not distinguishable using merely conditional independence tests. Also since the exogenous noises of both variables have changed, ordinary interventional tests are also not capable of using the information of these two environments to distinguish between the two structures [5]. Moreover, as it will be shortly explained (see Related Work), since the exogenous noise of the target variable has changed, the invariant prediction method [23], cannot discern the correct structure either. As another example, consider the structure in Figure 1(b). Suppose the exogenous noise of $X_1$ is varied across the two environments. Similar to the previous example, it can be shown that $\beta_{X_2}(X_1)$ varies across the two environments while $\beta_{X_1}(X_2)$ remains the same. This implies that the edge between $X_1$ and $X_2$ is from the former to the later. Similarly, $\beta_{X_3}(X_2)$ varies across the two environments while $\beta_{X_2}(X_3)$ remains the same. This implies that $X_2$ is the parent of $X_3$. Therefore, the structure in Figure 1(b) is distinguishable using the proposed identification approach. Note that the invariant prediction method cannot identify the relation between $X_2$ and $X_3$, and conditional independence tests are also not able to distinguish this structure.

**Related Work.** The main approach to learning the causal structure in observational setting is to run a constraint-based or a score-based algorithm over the data. Constraint-based approach [33, 21] is based on performing statistical tests to learn conditional independencies among the variables along with applying the Meek rules introduced in [36]. IC and IC* [21], PC, and FCI [33] algorithms are among the well known examples of this approach. In score-based approach, first a hypothesis space of potential models along with a scoring function is defined. The scoring function measures how well the model fits the observed data. Then the highest-scoring structure is chosen as the output (usually via greedy search). Greedy Equivalence Search (GES) algorithm [20, 2] is an example of score-based approach. Such purely observational approaches reconstruct the causal graph up to Markov equivalence classes. Thus, directions of some edges may remain unresolved. There are studies which attempt to identify the exact causal structure by restricting the model class [32, 12, 24, 22]. Most of such works consider SEM with independent noise. LiNGAM method [32] is a potent approach capable of structure learning in linear SEM model with additive noise[2], as long as the distribution of the noise is not Gaussian. Authors of [12] and [38] showed that a nonlinear SEM with additive noise,

and even the post-nonlinear causal model, along with some mild conditions on the functions and data distributions, are not symmetric in the cause and effect. There is also a line of work on causal structure learning in models where each vertex of the graph represents a random process [26, 34, 25, 6, 7, 16]. In such models, a temporal relationship is considered among the variables and it is usually assumed that there is no instantaneous influence among the processes. In interventional approach for causal structure learning, the experimenter picks specific variables and attempts to learn their relation with other variables, by observing the effect of perturbing that variables on the distribution of others. In recent works, bounds on the required number of interventions for complete discovery of causal relationships as well as passive and adaptive algorithms for minimizing the number of experiments were derived [5, 9, 10, 11, 31].

In this work we assume that the functional relations of the variables to their direct causes across a set of environments are invariant. Similar assumptions have been considered in other work [3, 30, 14, 13, 29, 23]. Specifically, [3] which studied finding causal relation between two variables related to each other by an invertible function, assumes that " the distribution of the cause and the function mapping cause to effect are independent since they correspond to independent mechanisms of nature".

There is little work on multi-environment setup [35, 23, 37]. In [35], the authors analyzed the classes of structures that are equivalent relative to a stream of distributions and presented algorithms that output graphical representations of these equivalence classes. They assumed that changing the distribution of a variable, varies the marginal distribution of all its descendants. Naturally this also assumes that they have access to enough samples to test each variable for marginal distribution change. This approach cannot identify the causal relations among variables which are affected by environment changes in the same way. The most closely related work to our approach is the invariant prediction method [23], which utilizes different environments to estimate the set of predictors of a target variable. In that work, it is assumed that the exogenous noise of the target variable does not vary among the environments. In fact, the method crucially relies on this assumption as it adds variables to the estimated predictors set only if they are necessary to keep the distribution of the target variable's noise fixed. Besides high computational complexity, invariant prediction framework may result in a set which does not contain all the parents of the target variable. Additionally, the optimal predictor set (output of the algorithm) is not necessarily unique. We will show that in many cases our proposed approach can overcome both these issues. Recently, the authors of [37] considered the setting in which changes in the mechanism of variables prevents ordinary conditional independence based algorithms from discovering the correct structure. The authors have modeled these changes as multiple environments and proposed a general solution for a non-parametric model which first detects the variables whose mechanism changed and then finds causal relations among variables using conditional independence tests. Due to the generality of the model, this method may require a high number of samples.

**Contribution.** We propose a novel causal structure learning framework, which is capable of uniquely identifying causal structures that were not identifiable using observational approaches, from information that was not usable in existing interventional approaches. The main contribution of this work is to introduce the idea of using the invariance of the functional relations of the variables to their direct causes across a set of environments. This would imply using the invariance of coefficients in the special case of linear SEM for distinguishing the causes from the effects. We define a notion of completeness for a causal inference algorithm in this setting and prove the existence of such algorithm by proposing the baseline algorithm (Section 3). Additionally, we present an alternate algorithm (Section 4) which has significantly improved computational and sample complexity compared to the baseline algorithm.

## 2 Regression-Based Causal Structure Learning

**Definition 1.** *Consider a directed graph $G = (V, E)$ with vertex set $V$ and set of directed edges $E$. $G$ is a DAG if it is a finite graph with no directed cycles. A DAG $G$ is called causal if its vertices represent random variables $V = \{X_1, ..., X_n\}$ and a directed edges $(X_i, X_j)$ indicates that variable $X_i$ is a direct cause of variable $X_j$.*

We consider a linear SEM [1] as the underlying data generating model. In such a model the value of each variable $X_j \in V$ is determined by a linear combination of the values of its causal parents $PA(X_j)$ plus an additive exogenous noise $N_j$ as follows

$$X_j = \sum_{X_i \in PA(X_j)} b_{ji} X_i + N_j, \qquad \forall j \in \{1, \cdots, p\}, \tag{1}$$

where $N_j$'s are jointly independent. This model could be represented by a single matrix equation $\mathbf{X} = \mathbf{B}\mathbf{X} + \mathbf{N}$. Further, we can write

$$\mathbf{X} = \mathbf{A}\mathbf{N}, \tag{2}$$

where $\mathbf{A} = (\mathbf{I} - \mathbf{B})^{-1}$. This implies that each variable $X \in V$ can be written as a linear combination of the exogenous noises in the system. We assume that in our model, all variables are observable. Also, we focus on zero-mean Gaussian exogenous noise; otherwise, the proposed approach could be extended to any arbitrary distribution for the exogenous noise in the system. The following definitions will be used throughout the paper.

**Definition 2.** *Graph union of a set $\mathcal{G}$ of mixed graphs[3] over a skeleton, is a mixed graph with the same skeleton as the members of $\mathcal{G}$ which contains directed edge $(X, Y)$, if $\exists\, G \in \mathcal{G}$ such that $(X, Y) \in E(G)$ and $\nexists\, G' \in \mathcal{G}$ such that $(Y, X) \in E(G')$. The rest of the edges remain undirected.*

**Definition 3.** *Causal DAGs $G_1$ and $G_2$ over $V$ are Markov equivalent if every distribution that is compatible with one of the graphs is also compatible with the other. Markov equivalence is an equivalence relationship over the set of all graphs over $V$ [17]. The graph union of all DAGs in the Markov equivalence class of a DAG $G$ is called the essential graph of $G$ and is denoted by $Ess(G)$.*

We consider a multi-environment setting consisting of $N$ environments $\mathcal{E} = \{E_1, ..., E_N\}$. The structure of the causal DAG and the functional relations for producing the variables from their parents (the matrix $\mathbf{B}$), remains the same across all environments, the exogenous noises may vary though. For a pair of environments $E_i, E_j \in \mathcal{E}$, let $I_{ij}$ be the set of variables whose exogenous noise have changed between the two environments. Given $I_{ij}$, for any DAG $G$ consistent with the essential graph[4] obtained from an observational algorithm, define the *regression invariance set* as follows

$$R(G, I_{ij}) \coloneqq \{(X, S) : X \in V, S \subseteq V \backslash \{X\}, \beta_S^{(i)}(X) = \beta_S^{(j)}(X)\},$$

where $\beta_S^{(i)}(X)$ and $\beta_S^{(j)}(X)$ are the regression coefficients of regressing variable $X$ on $S$ in environments $E_i$ and $E_j$, respectively. In words, $R(G, I_{ij})$ contains all pairs $(X, S)$, $X \in V$, $S \subseteq V \backslash \{X\}$ that if we regress $X$ on $S$, the regression coefficients do not change across $E_i$ and $E_j$.

**Definition 4.** *Given $I$, the set of variables whose exogenous noise has changed between two environments, DAGs $G_1$ and $G_2$ are called $I$-distinguishable if $R(G_1, I) \neq R(G_2, I)$.*

We make the following assumption on the distributions of the exogenous noises.

**Assumption 1** (Regression Stability Assumption)**.** *For a given set $I$ and structure $G$, there exists $\epsilon_0 > 0$ such that for all $0 < \epsilon \leq \epsilon_0$ perturbing the variance of the exogenous noises by $\epsilon$ does not change the regression invariance set $R(G, I)$.*

The purpose of Assumption 1 is to rule out pathological cases for values of the variance of the exogenous noises in two environments which make special regression relations. For instance, in Example 1, $\beta_{X_2}^{(1)}(X_1) = \beta_{X_2}^{(2)}(X_1)$ only if $\sigma_1^2 \tilde{\sigma}_2^2 = \sigma_2^2 \tilde{\sigma}_1^2$ where $\sigma_i^2$ and $\tilde{\sigma}_i^2$ are the variances of the exogenous noise of $X_i$ in the environments $E_1$ and $E_2$, respectively. Note that this special relation between $\sigma_1^2$, $\tilde{\sigma}_1^2$, $\sigma_2^2$, and $\tilde{\sigma}_2^2$ has Lebesgue measure zero in the set of all possible values for the variances. We give the following examples as applications of our approach.

**Example 1.** *Consider DAGs $G_1 : X_1 \rightarrow X_2$ and $G_2 : X_1 \leftarrow X_2$. For $I = \{X_1\}$, $I = \{X_2\}$ or $I = \{X_1, X_2\}$, calculating the regression coefficients as explained in Section 1, we see that $(X_1, \{X_2\}) \notin R(G_1, I)$ but $(X_1, \{X_2\}) \in R(G_2, I)$. Hence $G_1$ and $G_2$ are $I$-distinguishable. As mentioned in Section 1, structures $G_1$ and $G_2$ are not distinguishable using the observational tests. Also, in the case of $I = \{X_1, X_2\}$, the invariant prediction approach and the ordinary interventional tests - in which the experimenter expects that a change in the distribution of the effect would not perturb the marginal distribution of the cause variable - are not capable of distinguishing the two structures either.*

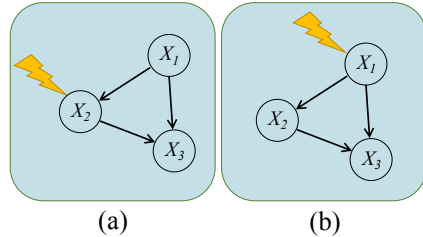

(a)       (b)

Figure 2: DAGs related to Example 3.

**Example 2.** *Consider the DAG $G$ in Figure 1(b) with $I = \{X_1\}$. Consider an alternative DAG $G'$ in which compared to $G$ the directed edge $(X_1, X_2)$ is replaced by $(X_2, X_1)$, and DAG $G''$ in which compared to $G$ the directed edge $(X_2, X_3)$ is replaced by $(X_3, X_2)$. Since $(X_2, \{X_1\}) \in R(G, I)$ while this pair is not in $R(G', I)$, and $(X_2, \{X_3\}) \notin R(G, I)$ while this pair belongs to $R(G'', I)$, the structure of $G$ is also distinguishable using the proposed identification approach. Note that the direction of the edges of $G$ is not distinguishable using an observational test as it has two other DAGs in its equivalence class. Also, the invariant prediction method cannot identify the relation between $X_2$ and $X_3$, since it can keep the variance of the noise of $X_3$ fixed by setting the predictor set as $\{X_2\}$ or $\{X_1\}$, which have empty intersection.*

**Example 3.** *Consider the structure in Figure 2(a) with $I = \{X_2\}$. Among the six possible triangle DAGs, all of them are $I$-distinguishable from this structure and hence, with two environments differing in the exogenous noise of $X_2$, this triangle DAG could be identified. Note that all the triangle DAGs are in the same Markov equivalence class and hence, using the information of one environment alone, observation only setting cannot lead to identification, which makes this structure challenging to deal with [8]. For $I = \{X_1\}$, the structure in Figure 2(b) is not $I$-distinguishable from a triangle DAG in which the direction of the edge $(X_2, X_3)$ is flipped. These two DAGs are also not distinguishable using the invariant prediction method and usual interventional approaches with intervention on $X_1$.*

Let the structure $G^*$ be the ground truth DAG structure. Define $\mathcal{G}(G^*, I) := \{G : R(G, I) = R(G^*, I)\}$, which is the set of all DAGs which are not $I$-distinguishable from $G^*$. Using this set, we form the mixed graph $M(G^*, I)$ over $V$ as the graph union of members of $\mathcal{G}(G^*, I)$.

**Definition 5.** *Let $P_i$ be the joint distribution over the set of variables $V$ in environment $E_i \in \mathcal{E}$. An algorithm $\mathscr{A} : (\{P_i\}_{i=1}^N) \to M$ which gets the joint distribution over $V$ in environments $\mathcal{E} = \{E_i\}_{i=1}^N$ as the input and returns a mixed graph, is regression invariance complete if for any pair of environments $E_i$ and $E_j$ with $I_{ij}$ as the set of variables whose exogenous noise has changed between $E_i$ and $E_j$, the set of directed edges of $M(G^*, I_{ij})$ be a subset of the set of directed edges of the output of $\mathscr{A}$.*

In Section 3 we will introduce a structure learning algorithm which is complete in the sense of Definition 5.

## 3 Existence of Complete Algorithms

In this section we show the existence of complete algorithm (in the sense of Definition 5) for learning the causal structure among a set of variables $V$ whose dynamics satisfy the SEM in (1). The pseudo-code of the algorithm is presented in Algorithm 1.

Suppose $G^*$ is the ground truth structure. The algorithm first runs a complete observational algorithm to obtain the essential graph $Ess(G^*)$. For each pair of environments $\{E_i, E_j\} \in \mathcal{E}$, first the algorithm calculates the regression coefficients $\beta_S^{(i)}(Y)$ and $\beta_S^{(j)}(Y)$, for all $Y \in V$ and $S \subseteq V \setminus \{Y\}$, and forms the regression invariance set $R_{ij}$, which contains the pairs $(Y, S)$ for which the regression coefficients did not change between $E_i$ and $E_j$. Note that ideally $R_{ij}$ is equal to $R(G^*, I_{ij})$. Next, using the function ChangeFinder$(\cdot)$, we discover the set $I_{ij}$ which is the set of variables whose exogenous noises have varied between the two environments $E_i$ and $E_j$. Then using the function ConsistantFinder$(\cdot)$, we find $\mathcal{G}_{ij}$ which is the set of all possible DAGs, $G$ that are consistent with $Ess(G^*)$ and $R(G, I_{ij}) = R_{ij}$. That is, this set is ideally equal to $\mathcal{G}(G^*, I_{ij})$. After taking the

---

**Algorithm 1** The Baseline Algorithm

**Input:** Joint distribution over $V$ in environments $\mathcal{E} = \{E_i\}_{i=1}^N$.
Obtain $Ess(G^*)$ by running a complete observational algorithm.
**for** each pair of environments $\{E_i, E_j\} \subseteq \mathcal{E}$ **do**
  Obtain $R_{ij} = \{(Y, S) : Y \in V, S \subseteq V \setminus \{Y\}, \beta_S^{(i)}(Y) = \beta_S^{(j)}(Y)\}$.
  $I_{ij} = ChangeFinder(E_i, E_j)$.
  $\mathcal{G}_{ij} = ConsistentFinder(Ess(G^*), R_{ij}, I_{ij})$.

  $M_{ij} = \bigcup_{G \in \mathcal{G}_{ij}} G$.
**end for**
$M_{\mathcal{E}} = \bigcup_{1 \leq i,j \leq N} M_{ij}$.
Apply Meek rules on $M_{\mathcal{E}}$ to get $\hat{M}$.
**Output:** Mixed graph $\hat{M}$.

---

union of graphs in $\mathcal{G}_{ij}$, we form $M_{ij}$, which is the mixed graph containing all causal relations distinguishable from the given regression information between the two environments. This graph is ideally equal to $M(G^*, I_{ij})$. After obtaining $M_{ij}$ for all pairs of environments, the algorithm forms a mixed graph $M_{\mathcal{E}}$ by taking graph union of $M_{ij}$'s. We apply the Meek rules on $M_{\mathcal{E}}$ to find all extra orientations and output $\hat{M}$. Since for each pair of environments we are searching over all DAGs, and we take the graph union of $M_{ij}$'s, the baseline algorithm is complete in the sense of Definition 5.

**Obtaining the set $R_{ij}$:** In this part, for a given significance level $\alpha$, we will show how the set $R_{ij}$ can be obtained to have total probability of false-rejection less than $\alpha$. For given $Y \in V$ and $S \subseteq V\backslash\{Y\}$ in the environments $E_i$ and $E_j$, we define the null hypothesis $H_{0,Y,S}^{ij}$ as follows:

$$H_{0,Y,S}^{ij} : \exists \beta \in \mathbb{R}^{|S|} \text{ such that } \beta_S^{(i)}(Y) = \beta \text{ and } \beta_S^{(j)}(Y) = \beta. \tag{3}$$

Let $\hat{\beta}_S^{(i)}(Y)$ and $\hat{\beta}_S^{(j)}(Y)$ be the estimations of $\beta_S^{(i)}(Y)$ and $\beta_S^{(j)}(Y)$, respectively, obtained using the ordinary least squares estimator, and define the test statistic

$$\hat{T} := (\hat{\beta}_S^{(i)}(Y) - \hat{\beta}_S^{(j)}(Y))^T (s_i^2 \hat{\Sigma}_i^{-1} + s_j^2 \hat{\Sigma}_j^{-1})^{-1} (\hat{\beta}_S^{(i)}(Y) - \hat{\beta}_S^{(j)}(Y))/|S|, \tag{4}$$

where $s_i^2$ and $s_j^2$ are unbiased estimates of variance of $Y - (X_S)^T \beta_S^{(i)}(Y)$ and $Y - (X_S)^T \beta_S^{(j)}(Y)$ in environments $E_i$ and $E_j$, respectively, and $\hat{\Sigma}_i$ and $\hat{\Sigma}_j$ are sample covariance matrices of $\mathbb{E}[X_S(X_S)^T]$ in environments $E_i$ and $E_j$, respectively. If the null hypothesis holds, then $\hat{T} \sim F(|S|, n - |S|)$, where $F(\cdot, \cdot)$ is the $F$-distribution (see supplementary material for details).

We set the p-value of our test to be less than $\alpha/(p \times (2^{p-1} - 1))$. Hence, by testing all null hypotheses $H_{0,Y,S}^{ij}$ for any $Y \in V$ and $S \subseteq V\backslash\{Y\}$, we can obtain the set $R_{ij}$ with total probability of false-rejection less than $\alpha$.

**Function *ChangeFinder*($\cdot$):** We use Lemma 1 to find the set $I_{ij}$.

**Lemma 1.** *Given environments $E_i$ and $E_j$, for a variable $Y \in V$, if $\mathbb{E}[(Y - (X_S)^T \beta_S^{(i)}(Y))^2 | E_i] \neq \mathbb{E}[(Y - (X_S)^T \beta_S^{(j)}(Y))^2 | E_j]$ for all $S \subseteq N(Y)$, where $N(Y)$ is the set of neighbors of $Y$, then the variance of exogenous noise $N_Y$ is changed between the two environments. Otherwise, the variance of $N_Y$ is unchanged.*

See the supplementary material for the proof.

Based on Lemma 1, for any variable $Y$, we try to find a set $S \subseteq N(Y)$ for which the variance of $Y - (X_S)^T \beta_S(Y)$ remains fixed between $E_i$ and $E_j$ by testing the following null hypothesis:

$$\bar{H}_{0,Y,S}^{ij} : \exists \sigma \in \mathbb{R} \text{ s.t. } \mathbb{E}[(Y - (X_S)^T \beta_S^{(i)}(Y))^2 | E_i] = \sigma^2 \text{ and } \mathbb{E}[(Y - (X_S)^T \beta_S^{(j)}(Y))^2 | E_j] = \sigma^2.$$

In order to test the above null hypothesis, we can compute the variance of $Y - (X_S)^T \beta_S^{(i)}$ in $E_i$ and $Y - (X_S)^T \beta_S^{(j)}$ in $E_j$ and test whether these variances are equal using an $F$-test. If the p-value of the test for the set $S$ is less than $\alpha/(p \times 2^\Delta)$, then we will reject the null hypothesis $\bar{H}_{0,Y,S}^{ij}$, where $\Delta$ is the maximum degree of the causal graph. If we reject all hypothesis tests $\bar{H}_{0,Y,S}^{ij}$ for all $S \in N(Y)$, then we will add $Y$ to set $I_{ij}$. Since we are performing at most $p \times 2^\Delta$ (for each variable, at most $2^\Delta$ tests), we can obtain the set $I_{ij}$ with total probability of false-rejection less than $\alpha$.

**Function *ConsistentFinder*($\cdot$):** Let $D_{st}$ be the set of all directed paths from variable $X_s$ to variable $X_t$. For any directed path $d \in D_{st}$, we define the weight of $d$ as $w_d := \Pi_{(u,v) \in d} b_{vu}$ where $b_{vu}$ are coefficients in (1). By this definition, it can be seen that the entry $(t, s)$ of matrix $\mathbf{A}$ in (2) is equal to $[\mathbf{A}]_{ts} = \sum_{d \in D_{st}} w_d$. Thus, the entries of matrix $\mathbf{A}$ are multivariate polynomials of entries of $\mathbf{B}$. Furthermore,

$$\beta_S^{(i)}(Y) = \mathbb{E}[X_S(X_S)^T | E_i]^{-1} \mathbb{E}[X_S Y | E_i] = (\mathbf{A}_S \mathbf{\Lambda}_i \mathbf{A}_S^T)^{-1} \mathbf{A}_S \mathbf{\Lambda}_i \mathbf{A}_Y^T, \tag{5}$$

where $\mathbf{A}_S$ and $\mathbf{A}_Y$ are the rows corresponding to set $S$ and $Y$ in matrix $\mathbf{A}$, respectively, and matrix $\mathbf{\Lambda}_i$ is a diagonal matrix where $[\mathbf{\Lambda}_i]_{kk} = \mathbb{E}[(N_k)^2 | E_i]$. Therefore, the entries of vector $\beta_S^{(i)}(Y)$ are rational functions of entries in $\mathbf{B}$ and $\mathbf{\Lambda}_i$. Hence, the entries of Jacobian matrix of $\beta_S^{(i)}(Y)$ with respect to the diagonal entries of $\mathbf{\Lambda}_i$ are also rational expression of these parameters.

In function *ConsistentFinder*($\cdot$), we select any directed graph $G$ consistent with $Ess(G^*)$ and set $b_{vu} = 0$ if $(u, v) \notin G$. In order to check whether $G$ is in $\mathcal{G}_{ij}$, we initially set $R(G, I_{ij}) = \emptyset$. Then, we compute the Jacobian matrix of $\beta_S^{(i)}(Y)$ parametrically for any $Y \in V$ and $S \in V\backslash\{Y\}$. As noted above, the entries of Jacobian matrix can be obtained as rational expressions of entries in $\mathbf{B}$ and $\mathbf{\Lambda}_i$. If all columns of Jacobian matrix corresponding to the elements of $I_{ij}$ are zero, $\beta_S^{(i)}(Y)$ is not changing by varying the variances of exogenous noises in $I_{ij}$ and hence, we add $(Y, S)$ to set $R(G, I_{ij})$. After checking all $Y \in V$ and $S \in V\backslash\{Y\}$, we add the graph $G$ in $\mathcal{G}_{ij}$ if $R(G, I_{ij}) = R_{ij}$.

---

**Algorithm 2** LRE Algorithm

---

**Input:** Joint distribution over $V$ in environments $\mathcal{E} = \{E_i\}_{i=1}^{N}$.

**Stage 1:** Obtain $Ess(G^*)$ by running a complete observational algorithm, and for all $X \in V$, form $PA(X)$, $CH(X)$, $UK(X)$.

**Stage 2:**

**for** each pair of environments $\{E_i, E_j\} \subseteq \mathcal{E}$ **do**

  **for** all $Y \in V$ **do**

    **for** each $X \in UK(Y)$ **do**

      Compute $\beta_X^{(i)}(Y)$, $\beta_X^{(j)}(Y)$, $\beta_Y^{(i)}(X)$, and $\beta_Y^{(j)}(X)$.

      **if** $\beta_X^{(i)}(Y) \neq \beta_X^{(j)}(Y)$, but $\beta_Y^{(i)}(X) = \beta_Y^{(j)}(X)$ **then**

        Set $X$ as a child of $Y$ and set $Y$ as a parent of $X$.

      **else if** $\beta_X^{(i)}(Y) = \beta_X^{(j)}(Y)$, but $\beta_Y^{(i)}(X) \neq \beta_Y^{(j)}(X)$ **then**

        Set $X$ as a parent of $Y$ and set $Y$ as a child of $X$.

      **else if** $\beta_X^{(i)}(Y) \neq \beta_X^{(j)}(Y)$, and $\beta_Y^{(i)}(X) \neq \beta_Y^{(j)}(X)$ **then**

        Find minimum set $S \subseteq N(Y) \backslash \{X\}$ such that $\beta_{S\cup\{X\}}^{(i)}(Y) = \beta_{S\cup\{X\}}^{(j)}(Y)$.

        **if** $S$ does not exist **then**

          Set $X$ as a child of $Y$ and set $Y$ as a parent of $X$.

        **else if** $\beta_S^{(i)}(Y) \neq \beta_S^{(j)}(Y)$ **then**

          $\forall W \in \{X\} \cup S$, set $W$ as a parent of $Y$ and set $Y$ as a child of $W$.

        **else**

          $\forall W \in S$, set $W$ as a parent of $Y$ and set $Y$ as a child of $W$.

        **end if**

      **end if**

    **end for**

  **end for**

**end for**

**Stage 3:** Apply Meek rules on the resulted mixed graph to obtain $\hat{M}$.

**Output:** Mixed graph $\hat{M}$.

---

## 4  LRE Algorithm

The baseline algorithm of Section 3 is presented to prove the existence of complete algorithms, but that algorithm is not practical due to its high computational and sample complexity. In this section we present the *Local Regression Examiner* (LRE) algorithm, which is an alternative much more efficient algorithm for learning the causal structure among a set of variables $V$. The pseudo-code of the algorithm is presented in Algorithm 2. We make use of the following result in this algorithm.

**Lemma 2.** *Consider adjacent variables $X, Y \in V$ in causal structure $G$. For a pair of environments $E_i$ and $E_j$, if $(X, \{Y\}) \in R(G, I_{ij})$, but $(Y, \{X\}) \notin R(G, I_{ij})$, then $Y$ is a parent of $X$.*

See the supplementary material for the proof.

LRE algorithm consists of three stages. In the first stage, similar to the baseline algorithm, it runs a complete observational algorithm to obtain the essential graph. Then for each variable $X \in V$, it forms the set of $X$'s discovered parents $PA(X)$, and discovered children $CH(X)$, and leaves the remaining neighbors as unknown in $UK(X)$. In the second stage, the goal is that for each variable $Y \in V$, we find $Y$'s relation with its neighbors in $UK(Y)$ based on the invariance of its regression on its neighbors across each pair of environments. To do so, for each pair of environments, after fixing a target variable $Y$ and for each of its neighbors in $UK(Y)$, the regression coefficients of $X$ on $Y$ and $Y$ on $X$ are calculated. We will face one of the following cases:

- If neither is changing, we do not make any decisions about the relationship of $X$ and $Y$. This case is similar to having only one environment, similar to the setup in [32].
- If one is changing and the other is unchanged, Lemma 2 implies that the variable which fixes the coefficient as the regressor is the parent.
- If both are changing, we look for an auxiliary set $S$ among $Y$'s neighbors with minimum number of elements, for which $\beta_{S\cup\{X\}}^{(i)}(Y) = \beta_{S\cup\{X\}}^{(j)}(Y)$. If no such $S$ is found, it implies that $X$ is a child of $Y$. Otherwise, if $S$ and $X$ are both required in the regressors set to fix the coefficient, we set $\{X\} \cup S$ as parents of $Y$; otherwise, if $X$ is not required in the regressors set to fix the

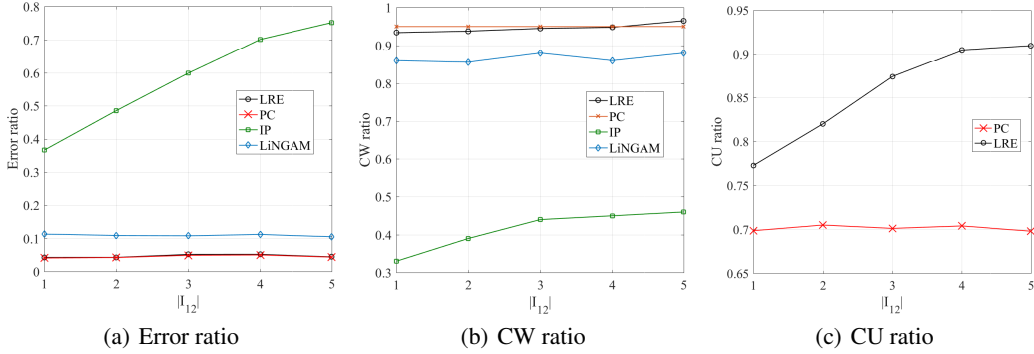

| (a) Error ratio | (b) CW ratio | (c) CU ratio |

Figure 3: Comparsion of performance of LRE, PC, IP, and LiNGAM algorithms.

coefficient, although we still set $S$ as parents of $Y$, we do not make any decisions regarding the relationship between $X$ and $Y$ (Example 3 when $I = \{X_1\}$, is an instance of this case).

After adding the discovered relationships to the initial mixed graph, in the third stage, we apply the Meek rules on the resulting mixed graph to find all extra possible orientations and output $\hat{M}$.

**Analysis of LRE Algorithm.** We can use the hypothesis testing in (3) to test whether two vectors $\beta_S^{(i)}(Y)$ and $\beta_S^{(j)}(Y)$ are equal for any $Y \in V$ and $S \subseteq N(Y)$. If the p-value for the set $S$ is less than $\alpha/(p \times (2^\Delta - 1))$, then we will reject the null hypothesis $H_{0,Y,S}^{ij}$. By doing so, we obtain the output with total probability of false-rejection less than $\alpha$. Regarding the computational complexity, since for each pair of environments, in the worse case we perform $\Delta(2^\Delta - 1)$ hypothesis tests for each variable $Y \in V$, and considering that we have $\binom{N}{2}$ pairs of environments, the computational complexity of LRE algorithm is in the order of $\binom{N}{2}p\Delta(2^\Delta - 1)$. Therefore, the bottleneck in the complexity of LRE is the requirement of running a complete observational algorithm in its first stage.

## 5 Experiments

We evaluate the performance of LRE algorithm by testing it on both synthetic and real data. As seen in the pseudo-code in Algorithm 2, LRE has three stages where in the first stage, a complete observational algorithm is run. In our simulations, we used the PC algorithm[5] [33], which is known to have a complexity of order $O(p^\Delta)$ when applied to a graph of order $p$ with degree bound $\Delta$.

**Synthetic Data.** We generated 100 DAGs of order $p = 10$ by first selecting a causal order for variables and then connecting each pair of variables with probability 0.25. We generated data from a linear Gaussian SEM with coefficients drawn uniformly at random from $[0.1, 2]$, and the variance of each exogenous noise was drawn uniformly at random from $[0.1, 4]$. For each variable of each structure, $10^5$ samples were generated. In our simulation, we only considered a scenario in which we have two environments $E_1$ and $E_2$, where in the second environment, the exogenous noise of $|I_{12}|$ variables were varied. The perturbed variables were chosen uniformly at random.

Figure 3 shows the performance of LRE algorithm. Define a link to be any directed or undirected edge. The error ratio is calculated as follows: *Error ratio* $:= (|miss\text{-}detected\ links| + |extra\ detected\ links| + |correctly\ detected\ wrongly\ directed\ edges|)/\binom{p}{2}$. Among the correctly detected links, define $C := |correctly\ directed\ edges|$, $W := |wrongly\ directed\ edges|$, and $U := |undirected\ edges|$. CW and DU ratios, are obtained as follows: *CW ratio* $:= (C)/(C + W)$, *CU ratio* $:= (C)/(C + U)$. As seen in Figure 3, only one change in the second environment (i.e., $|I_{12}| = 1$), increases the CU ratio of LRE by 8 percent compared to the PC algorithm. Also, the main source of error in LRE algorithm results from the application of the PC algorithm. We also compared the Error ratio and CW ratio of LRE algorithm with the Invariant Prediction (IP) [23] and LiNGAM [32] (since there is no undirected edges in the output of IP and LiNGAM, the CU ratio of both would be one). For LiNGAM, we combined the data from two environments as the input. Therefore, the distribution of the exogenous noise of variables in $I_{12}$ is not Guassian anymore. As it can be seen in Figure 3(a), the Error ratio of IP increases as the size of $I_{12}$ increases. This is mainly due to the fact that in IP approach it is assumed that the distribution of exogenous noise of the target variable should not change, which may be violated by increasing $|I_{12}|$. The result of simulations shows that the Error ratio of LiNGAM is

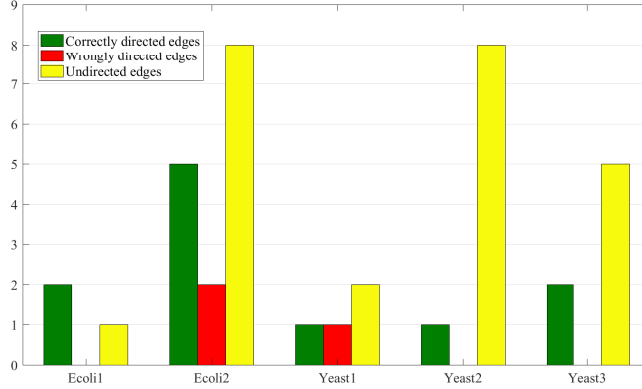

Figure 4: Performance of LRE algorithm in GRNs from DREAM 3 challenge. All five networks have 10 genes and total number of edges in each network (from left to right) is 11, 15, 10, 25, and 22, respectively.

approximately twice of those of LRE and PC. We also see that LRE performed better compared to LiNGAM and IP in terms of CW ratio.

**Real Data**
**a)** We considered dataset of educational attainment of teenagers [27]. The dataset was collected from 4739 pupils from about 1100 US high schools with 13 attributes including gender, race, base year composite test score, family income, whether the parent attended college, and county unemployment rate. We split the dataset into two parts where the first part includes data from all pupils who live closer than 10 miles to some 4-year college. In our experiment, we tried to identify the potential causes that influence the years of education the pupils received. We ran LRE algorithm on the two parts of data as two environments with a significance level of 0.01 and obtained the following attributes as a possible set of parents of the target variable: base year composite test score, whether father was a college graduate, race, and whether school was in urban area. The IP method [23] also showed that the first two attributes have significant effects on the target variable.

**b)** We evaluated the performance of LRE algorithm in gene regulatory networks (GRN). GRN is a collection of biological regulators that interact with each other. In GRN, the transcription factors are the main players to activate genes. The interactions between transcription factors and regulated genes in a species genome can be presented by a directed graph. In this graph, links are drawn whenever a transcription factor regulates a gene's expression. Moreover, some of vertices have both functions, i.e., are both transcription factor and regulated gene.
We considered GRNs in "DREAM 3 In Silico Network" challenge, conducted in 2008 [19]. The networks in this challenge were extracted from known biological interaction networks. The structures of these networks are available in the open-source tool "GeneNetWeaver (GNW)" [28]. Since we knew the true causal structures in these GRNs, we obtained $Ess(G^*)$ and gave it as an input to LRE algorithm. Furthermore, we used GNW tool to get 10000 measurements of steady state levels for every gene in the networks. In order to obtain measurements from the second environment, we increased coefficients of exogenous noise terms from 0.05 to 0.2 in GNW tool. Figure 4 depicts the performance of LRE algorithm in five networks extracted from GRNs of E-coli and Yeast bacteria. The green, red, and yellow bar for each network shows the number of correctly directed edges, wrongly directed edges, and undirected edges, respectively. Note that since we know the correct $Ess(G^*)$, there is no miss-detected links or extra detected links. As it can be seen, LRE algorithm has a fairly good accuracy (84% on average over all five networks) when it decides to orient an edge.

## 6  Conclusion

We studied the problem of causal structure learning in a multi-environment setting, in which the functional relations for producing the variables from their parents remain the same across environments, while the distribution of exogenous noises may vary. We defined a notion of completeness for a causal discovery algorithm in this setting and proved the existence of such algorithm. We proposed an efficient algorithm with low computational and sample complexity and evaluated the performance of this algorithm by testing it on synthetic and real data. The results show the efficacy of the proposed algorithm.

**Acknowledgments**

This work was supported in part by ARO grant W911NF-15-1-0281 and ONR grant W911NF-15-1-0479. Also, KZ acknowledges the support from NIH-1R01EB022858-01 FAIN-R01EB022858, NIH-1R01LM012087, and NIH-5U54HG008540-02 FAINU54HG008540. The content is solely the responsibility of the authors and does not necessarily represent the official views of the NIH.

## Footnotes

[1]As noted in [12], "nonlinearities can play a role similar to that of non-Gaussianity", and both lead to exact structure recovery.

[2] There are extensions to LiNGAM beyond linear model [38].

[3]A mixed graph contains both directed and undirected edges.

[4]DAG $G$ is consistent with mixed graph $M$, if they have the same skeleton and $G$ does not contain edge $(X, Y)$ while $M$ contains $(Y, X)$.

[5]We use the **pcalg** package [15] to run the PC algorithm on a set of random variables.

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
