[Supplementary Material · supplementary material.pdf]

# Appendices

## A  Test Statistic in Equation (4)

The null hypothesis $H_{0,Y,S}^{ij}$ can be written in the following matrix form: $\mathbf{C}\beta = 0$ where $\mathbf{C}$ is a $|S| \times (2|S|)$ matrix such that non-zero entries of $\mathbf{C}$ are $[\mathbf{C}]_{k,k} = 1$, $[\mathbf{C}]_{k,k+|S|} = -1$, for all $1 \leq k \leq |S|$, and $\beta$ is a $(2|S|) \times 1$ vector which is equal to $\begin{bmatrix} \beta_S^{(i)}(Y) \\ \beta_S^{(j)}(Y) \end{bmatrix}$.

The hypothesis tests in the form of $\mathbf{C}\beta = 0$ can be performed by $F$-tests (see Section 3.6 and Appendix C.7 in [18]). In particular, for the null hypothesis $H_{0,Y,S}^{ij}$, the following statistic

$$(\hat{\beta}_S^{(i)}(Y) - \hat{\beta}_S^{(j)}(Y))^T(\mathbf{C}\hat{\Sigma}\mathbf{C}^T)^{-1}(\hat{\beta}_S^{(i)}(Y) - \hat{\beta}_S^{(j)}(Y))/|S| \tag{6}$$

has a $F(|S|, n - |S|)$ distribution where $\hat{\Sigma} = [s_i^2\hat{\Sigma}_i^{-1}, \mathbf{0}_{|S|\times|S|}; \mathbf{0}_{|S|\times|S|}, s_j^2\hat{\Sigma}_j^{-1}]$. Since $\mathbf{C}\hat{\Sigma}\mathbf{C}^T = s_i^2\hat{\Sigma}_i^{-1} + s_j^2\hat{\Sigma}_j^{-1}$, the above statistic is equal to (4).

## B  Proof of Lemma 1

In a given environment $E_i$, for any set $S \subseteq N(Y)$, using representation (2), we have:

$$Y = \sum_{k:X_k \in AN(Y)\backslash\{Y\}} c_k N_k + N_Y,$$

$$X_S \beta_S^{(i)}(Y) = \sum_{k:X_k \in AN(Y)\backslash\{Y\}} b_k^{(i)} N_k + \sum_{k:X_k \in AN(S_{CH})\backslash AN(Y)} b_k'^{(i)} N_k + b_Y^{(i)} N_Y,$$

where $S_{CH} := S \cap CH(Y)$ and the ancestral set $AN(X)$ of a variable $X$ consists of $X$ and all the ancestors of nodes in $X$. Therefore

$$Y - X_S \beta_S^{(i)}(Y) = \sum_{k:X_k \in AN(Y)\backslash\{Y\}} (c_k - b_k^{(i)}) N_k - \sum_{k:X_k \in AN(S_{CH})\backslash AN(Y)} b_k'^{(i)} N_k + (1 - b_Y^{(i)}) N_Y,$$

If the variance of $N_Y$ is not changed, then for the choice of $S = PA(Y)$, the second summation vanishes, and in the first summation, we have: $c_k = b_k^{(i)}$ for all $X_k \in AN(Y)\backslash\{Y\}$ and $b_Y^{(i)} = 0$ due to regressing $Y$ on its parents. Therefore, the variance of residual remains unvaried. Otherwise, if the variance of $N_Y$ changes across two environments, then this change may cancel out only for specific values of the variances of other exogenous noises, which according to a similar reasoning as the one in Assumption 1, we assume that this case does not happen.

## C  Proof of Lemma 2

Suppose $X$ is the parent of $Y$. Consider environments $E_i, E_j \in \mathcal{E}$. It suffices to show that if $\beta_Y^{(i)}(X) = \beta_Y^{(j)}(X)$, then $\beta_X^{(i)}(Y) = \beta_X^{(j)}(Y)$. Using representation (2), $X$ and $Y$ can be expressed as follows

$$X = \sum_{k:X_k \in AN(X)} a_k N_k,$$

$$Y = \sum_{k:X_k \in AN(X)} b_k N_k + \sum_{k:X_k \in AN(Y)\backslash AN(X)} c_k N_k.$$

Hence we have

$$\mathbb{E}[X^2] = \sum_{k:X_k \in AN(X)} a_k^2 var(N_k),$$

$$\mathbb{E}[Y^2] = \sum_{k:X_k \in AN(X)} b_k^2 var(N_k) + \sum_{k:X_k \in AN(Y) \backslash AN(X)} c_k^2 var(N_k),$$

$$\mathbb{E}[XY] = \sum_{k:X_k \in AN(X)} a_k b_k var(N_k).$$

Therefore

$$\beta_X(Y) = \frac{\sum_{k:X_k \in AN(X)} a_k b_k var(N_k)}{\sum_{k:X_k \in AN(X)} a_k^2 var(N_k)}$$

$$\beta_Y(X) = \frac{\sum_{k:X_k \in AN(X)} a_k b_k var(N_k)}{\sum_{k:X_k \in AN(X)} b_k^2 var(N_k) + \sum_{k:X_k \in AN(Y) \backslash AN(X)} c_k^2 var(N_k)}$$

in the expression for $\beta_Y(X)$, the first summation contains the same exogenous noises as the numerator while the second summation contains terms related to the variance of other orthogonal exogenous noises. Therefore, by Assumption 1, $\beta_Y^{(i)}(X) = \beta_Y^{(j)}(X)$ only if for all $k : X_k \in AN(Y)$, $var(N_k)$ remains unchanged. In this case, we will also have $\beta_X^{(i)}(Y) = \beta_X^{(j)}(Y)$. Note that $\beta_X(Y)$ can always remain unchanged if the exogenous noise of variables in $AN(X)$ affect $Y$ only through $X$.