[Reviews · NeurIPS 2017]

Reviewer 1



The authors propose using regression systematically in a causal setup where an intervention refers to a change in the distribution of some of the exogenous variables. They propose two algorithms for this, one is complete but computationally intractable, whereas the other is a heuristic. One of the most important problems is the motivation for this new type of intervention. Please provide real world examples of when this type of intervention is realistic. Another important problem is about Assumption 1. This assumption is very convenient for the analysis but it is very unclear what it entails. How should the variances be related to one another for Assumption 1 to hold? And under what perturbation? Authors should provide a more detailed alternate assumption instead that clearly lays out how noise variances relate to each other, while also considering the causal graph. Otherwise, it is very difficult to assess how restrictive this assumption is. Line 38: "Note that this model is one of the most problematic models in the literature of causal inference" -> Additive noise models with Gaussian exogenous variables are arguably one of the most restrictive models. It is not clear if the hardness of identification in this case is relevant for practical questions. Line 62, 63, 64: "Also since the exogenous noises of both variables have changed, commonly used interventional tests are also not capable of distinguishing between these two structures [4]" This is not true. As long as the noise variables are independent (they are always independent in Pearlian setting under causal sufficiency), a simple conditional independence test in the two environments, one pre-interventional and one post-interventional, yields the true causal direction (assuming either X1 -> X2 or X1 <- X2). The fact that exogenous variables have different distributions does not disturb this test: Simply observe X1 and X2 are independent under an intervention on X2 to conclude X1->X2. Line 169: "ordinary interventional tests" are described as the changes observed in the distribution of effect given a perturbation of the cause distribution. However, the more robust interventional test commonly used in the literature checks the conditional independence of the two variables in the post-interventional distribution, as explained in the above comment. Please see "Characterization and Greedy Learning of Interventional Equivalence Classes" by Buhlmann for details. Based on the two comments above, please do not include "discovering causal graph where interventional methods cannot" in the list of contributions, this is incorrect and severely misleading. I believe the authors should simply say that methods used under perfect interventions cannot be used any more under this type of intervention - which changes the distribution of the exogenous variable - since no new conditional independencies are created compared to the observational distribution. This would be the correct statement, but it is not what the paper currently reads. Please edit the misleading parts of the text. Example 1 is slightly confusing: Authors use I = {X1} which does not have any subscript as the previous notation I_{i,j}. I believe it would be more clear to write it as I = {\emptyset, X1} instead. It is not clear if the algorithm LRE is complete, I believe not. Please explicitly mention this in the paper. The regression invariance set - R - can be indicative of the set of parents of a variable. Authors already make use of this observation in designing the LRE algorithm. What if every exogenous variable changed across every two intervention, i.e., I_{i,j} = [n]. How would you change LRE algorithm for this special case. I would intuitively believe it would greatly simplify it. This may be a useful special case to consider that can boost the usefulness of regression invariant sets. Another extension idea for the paper is to design the experiments I_{i,j}. What is the minimum number of interventions required? Simulation results are very preliminary. The results of the real dataset simulations is not clear: Authors have identified a set of possible causes of the number of years of education. Since the true causal graph is unknown it is not clear how to evaluate the results.

Reviewer 2



The paper studies the task of inferring causal DAGs in a scenario where the structural equations are constant across different data sets, while the unobserved noise terms change. The approach is interesting, relevant and sound. The presentation is ok, but could be improved. For instance, the description of invariant prediction must be more explicit -- what is assumed to change and what is assumed to remain constant? In particular, I didn't see why it fails to identify the DAG in Fig 1b due to the lack of clarity about the invariant prediction scenario. Some remarks / questions: - Some readers may appreciate a remark about why it is natural that the noise variable changes while the structure coefficient remains the same. After all, both are just parameters of the conditional distribution of a variable given its causes. - Assumption 1: 'small value \epsilon' is not defined. Maybe say s.th. like 'if there is an epsilon_0 such that R(G,I) is not changed for perturbations \epsilon smaller than \epsilon_0'. - Ref. [2] appeared at UAI 2010 - In line 56, the authors exclude "pathological cases", which seems to exclude that two noise terms change jointly in a way that ensures that non-generic regression invariances occur. It would be nice to be more explicit about this and postulate such an independence assumption rather than stating it implicitly (unless I didn't get you right). - I'm wondering to what extent the additional causal information provided by regression invariance could also be captured by a scenario where one introduces an additional variable labelling the data set. I'm not saying that this would be equivalent to your scenario, it would just be nice to see a few remarks why yours is superior. - I missed the point why it's important that the Jacobian matrix mentioned in line 268 and 272 is a rational function of the entries of B - In line 238 'Furthermore, we have \Sigma_i =...' --> this is the *definition* of \Sigma_i, right? Please indicate. - Appendix A: I didn't get the meaning of the notation C[\beta .... ; ...]. Since there are no space constraints, these statements could be properly explained. - Please indicate what F(p,n-p) stands for - Would non-linear structural equations yield stronger identifiability results?